# Systematic analysis of proton pump inhibitors-related adverse reactions using the FDA adverse event reporting system database

Zhenyu Wang[1,2], Jianan Jin [3], Guimei Wang[1], Guoqi Zhou[4]*, Hanliang Jiang[1]*

**1** Pulmonary and Critical Care Medicine, Regional medical center for National Institute of Respiratory Disease, Sir Run Run Shaw Hospital, School of Medicine, Zhejiang University, Hangzhou, Zhejiang, P. R. China, **2** School of Medicine, Shaoxing university, Shaoxing, Zhejiang, People's Republic of China, **3** Graduate School, Zhejiang Chinese Medical University, People's Republic of China, **4** Pulmonary Disease Department, Zunyi of Traditional Hospital Chinese Medicine, Zunyi, Guizhou, People's Republic of China

* aock@zju.edu.cn (HJ); 31343639@qq.com (GZ)

## Abstract

### Background

Proton pump inhibitors (PPIs), known for their potent acid-suppressing effects, are widely used in various clinical settings, including treatment and prevention. Understanding their adverse effects is crucial. This study, utilizing the FDA Adverse Event Reporting System (FAERS) database, comprehensively analyzes PPI-related adverse events to guide clinical medication practices.

### Methods

This study analyzed suspected adverse drug reactions (ADRs) related to specific PPI drugs using data from the FAERS database, covering Q1 2004 to Q4 2024. Multiple statistical methods, including ROR, PRR, IC025, and EBGM, were employed for evaluation, with ADRs defined according to System Organ Class (SOC) and Preferred Term (PT). A comparative analysis was conducted to assess potential differences in ADR profiles among different PPI drugs.

### Results

This study analyzed 176,680 cases of PPI-related adverse events, with a total of 632,468 adverse reaction reports recorded when PPIs were designated as the primary suspected drug (PS). PPIs showed significantly elevated risks in the renal/urinary and gastrointestinal systems, with other common adverse reactions including hypomagnesemia, hypocalcemia, and renal anemia. Most adverse reactions occurred either within the first 0–30 days of use or after prolonged exposure (>6 months), and elderly patients (≥65 years) were disproportionately affected.

**Data availability statement:** All relevant data are within the paper and its Supporting Information files.

**Funding:** This work was supported by the National Natural Science Foundation of China (Grant No. 62176230) and the Zhejiang Medical and Health Science and Technology Project (Grant No. 2022KY827). The funders had no role in study design, data collection and analysis, decision to publish, or preparation of the manuscript.

**Competing interests:** The authors have declared that no competing interests exist.

## Conclusions

For high-risk populations using PPIs long-term (such as elderly patients or those with pre-existing renal impairment), continuous monitoring is essential to mitigate potential complications. Unnecessary use should be strictly avoided, and long-term medication should be minimized to ensure safety and appropriateness.

## 1. Introduction

PPIs are among the most widely prescribed medications globally. Their potent acid-suppressive effects make them a mainstay in the management of acid-related disorders such as gastroesophageal reflux disease (GERD), peptic ulcer disease (PUD), and Helicobacter pylori eradication therapy [1,2]. Furthermore, PPIs are extensively utilized for preventing gastrointestinal complications associated with long-term use of certain medications, including nonsteroidal anti-inflammatory drugs (NSAIDs), anticoagulants, and corticosteroids [3,4]. Studies have consistently documented a high prevalence of off-label PPI use worldwide, encompassing both inappropriate utilization without valid indications and prolonged maintenance therapy without dose reduction despite established indications. Although post-marketing surveillance generally supports the safety and efficacy of PPIs as acid-suppressive agents, an increasing number of adverse events related to their long-term administration have been reported [5–7]. Growing research attention is now directed toward potential safety concerns associated with chronic PPI use. Evidence from case reports and cohort studies continues to deepen our understanding of both the clinical manifestations and underlying mechanisms of PPI-associated adverse reactions.

PPIs exert their therapeutic effects by irreversibly inhibiting the $H^+/K^+$-ATPase enzyme in parietal cells, thereby substantially reducing gastric acid secretion. This mechanism not only alleviates symptoms of acid-related disorders but also promotes gastric mucosal healing, solidifying their pivotal role in gastroenterological therapeutics. However, the pharmacodynamic action of acid suppression carries inherent risks [8]. Prolonged gastric acid inhibition may disrupt gastrointestinal microbiome homeostasis, impair nutrient absorption, and interfere with other physiological processes, potentially leading to adverse outcomes such as renal injury, hypomagnesemia, increased fracture risk, heightened susceptibility to infections, and cardiovascular events [9,10]. These multifaceted adverse effects have been extensively investigated and reported, further amplifying concerns about PPI overuse and underscoring the imperative for judicious clinical application [11].

Pharmacovigilance databases possess unique advantages in post-marketing drug safety surveillance. These systems enable timely detection of medication-related safety issues through continuous monitoring, thereby mitigating potential risks. While previous studies utilizing FDA databases have analyzed ADRs associated with specific PPIs – such as Chen et al.'s investigation highlighting concerns regarding PPI use in patients predisposed to acute kidney injury (AKI) [12], Zhai's identification of PPIs potentially inducing various cardiac and vascular events (CVEs) [13], Sun's

detection of significant rhabdomyolysis signals [14], and Zhang's exploration of potential oncogenic risks – the absence of standardized ADR assessment criteria has limited their ability to provide comprehensive comparative analyses [15]. The growing demand for precision medicine and personalized therapy underscores the critical need for holistic understanding of these pharmacological variations, a gap our research aims to address. Furthermore, the continuous accumulation of adverse event data necessitates updated investigations to elucidate emerging ADR patterns and validate previous findings. Consequently, systematic characterization of PPI-associated adverse reactions and their underlying mechanisms is imperative for optimizing clinical pharmacotherapy.

This study conducted a systematic retrospective analysis of PPI-associated adverse reactions using data mining techniques applied to the FAERS database. Through quantitative disproportionality analyses including reporting odds ratio (ROR), proportional reporting ratio (PRR), information component (IC025), and empirical Bayesian geometric mean (EBGM), we investigated the potential risks and associations of PPI-related ADRs. Furthermore, comparative analyses of inter-PPI pharmacovigilance profiles were performed to elucidate class-specific versus agent-specific safety concerns, thereby providing evidence-based insights for optimizing therapeutic decision-making and enhancing medication safety in clinical practice.

## 2. Materials and methods

### 2.1. Data source

This study employed the FAERS database, which contains over 20 million ADR reports from the United States, Europe, and Asia. Reports are submitted by both healthcare professionals (including physicians, pharmacists, and registered nurses) and non-healthcare individuals (such as consumers, lawyers, and vendors). Each report provides a unique case identifier, along with patient demographic details, report date, reporting country, qualifications of the primary reporter, suspected drugs and their indications, ADR occurrence date, severity level, and detailed ADR descriptions. To standardize ADR reporting, each event is coded using PT from the Medical Dictionary for Regulatory Activities (MedDRA) 27.0, with classifications organized according to SOC.

PPIs were approved for clinical use in the late 1980s, and have since experienced significant global market expansion. This study adopted a retrospective pharmacovigilance study design and analyzed FAERS-reported ADRs related to PPIs from Q1 2004 to Q4 2024, covering six active ingredients: esomeprazole, lansoprazole, omeprazole, pantoprazole, rabeprazole, and dexlansoprazole. Both brand and generic names were included, and data underwent deduplication and filtering following FDA recommendations. Data cleaning followed a standardized deduplication rule recommended by the FDA: reports were sorted using CASEID, FDA_DT, and PRIMARYID from the DEMO table. Among reports with identical CASEID, the one with the latest FDA_DT was retained. If both CASEID and FDA_DT were identical, the report with the highest PRIMARYID was preserved. Further cleaning was applied by excluding records with missing or implausible values for key variables such as age (0–120 years), gender (only "Male" or "Female"), and weight (20–200 kg), resulting in a finalized cleaned dataset. To minimize potential confounding from concomitant medications, combination therapies, such as PPIs co-administered with antibiotics, were excluded to reduce false-positive signals. Additionally, reports involving non-drug-related factors, including procedural complications, product issues, and socio-environmental influences, were removed to ensure that only genuine ADR signals were retained. Two independent researchers conducted data processing and validation. In cases of discrepancy, a third researcher reviewed the data and provided the final determination. A targeted literature review was also conducted to contextualize the findings and compare with existing evidence on renal adverse effects of PPIs.

### 2.2. Data analysis

Following data cleaning and deduplication, a total of 176,680 cases involving PPI-related adverse events were included in this study. The analysis covered six major PPIs: Esomeprazole, Lansoprazole, Dexlansoprazole, Omeprazole,

Pantoprazole, and Rabeprazole. When PPIs were designated as the primary suspect drug (PS), a total of 632,468 adverse reaction reports were identified. To assess the disproportionate reporting of adverse events, we conducted disproportionality analysis, a widely used method in pharmacovigilance. The following statistical approaches were applied [16,17]:

ROR measures the strength of association between a drug and an adverse event by comparing the frequency of the event in the target drug group to its frequency in the remaining dataset. A ROR lower limit (ROR025) exceeding 1 is considered a positive signal. PRR compares the proportion of reports for a specific adverse event associated with a drug against the proportion for all other drugs. A PRR ≥ 2, $\chi^2$ ≥ 4, and a minimum of 3 cases indicate a potential signal. Information Component (IC025) – Based on Bayesian statistics, IC quantifies the deviation of observed event frequencies from the expected values. A lower limit of the 95% confidence interval (IC025) > 0 is considered a significant signal. EBGM is a Bayesian method that smooths probability estimates to control for small sample sizes. A lower bound (EB05) exceeding 1 is indicative of a signal.

To ensure the robustness of signal detection, an adverse event was considered a true positive signal only if all four methods yielded positive results. All statistical analyses were conducted using Excel (Microsoft Office 365) and R (version 4.4.2) for data processing, visualization, and statistical calculations.

## 3. Results

### 3.1. Study population

From Q1 2004 to Q4 2024, after data cleaning and deduplication, a total of 176,680 cases of PPIs were reported as the primary PS were included in this study, encompassing 632,468 adverse events. Among these, serious adverse events accounted for 71.8% (n = 126,945), while fatal cases constituted 6.2% (n = 11,005). After excluding cases with missing demographic information, female patients (n = 88,745, 50.2%) accounted for a significantly higher proportion of ADR reports than males (n = 54,313, 30.7%). The age distribution was predominantly middle-aged (45–64 years, 24.4%) and elderly (≥65 years, 25.3%). Geographically, the United States contributed the highest proportion of cases (68.18%), with consumers being the primary reporters (38.6%). Detailed information is presented in Table 1.

As illustrated in (Fig 1), the number of adverse event reports associated with six PPIs has shown a consistent upward trend over the past two decades (2004–2024). Among these, Dexlansoprazole was approved by the FDA in 2009, and its adverse event reports have been recorded since then. The other five PPIs reached their peak reporting volume in 2019, whereas Rabeprazole peaked earlier in 2018. Esomeprazole exhibited a secondary peak in 2012. This increasing trend may be attributed to multiple factors, including the continuous improvement of the FAERS database, the strengthening of pharmacovigilance systems, the growing prescription volume of PPIs, and the widespread availability of OTC formulations. Additionally, heightened public awareness regarding potential PPI-related adverse events and the intensification of drug safety monitoring have likely further propelled the rise in reported cases.

### 3.2. Primary analysis

The adverse event reporting pattern associated with PPIs exhibits a distinct time-dependent trend. As shown in (Fig 2), the highest incidence of adverse event reports occurs within the first 0–30 days of PPI use, accounting for more than half of the total reports. This suggests that the risk of adverse reactions is particularly elevated during the initial phase of treatment, necessitating heightened clinical vigilance. In the initial stages of therapy, adjustments should be made promptly based on patient feedback to minimize the occurrence of adverse events. Over time, the reporting rate gradually declines. However, a resurgence in adverse event reports is observed after six months (>180 days), culminating in a second peak during long-term use (>360 days). This pattern highlights the potential for cumulative effects and delayed-onset adverse reactions with prolonged PPI therapy, emphasizing the need for continuous monitoring throughout treatment to mitigate long-term risks.

 

**Table 1. Basic characteristics of PPIs in FAERS.**

| Characteristic | Esomeprazole (%) | Lansoprazole (%) | Dexlansopra-zole (%) | Omeprazole (%) | Pantoprazole (%) | Rabeprazole (%) | Total |
|---|---|---|---|---|---|---|---|
| Total cases | 69438 | 27461 | 5882 | 38783 | 31216 | 3900 | 176680 |
| **Gender** | | | | | | | |
| Female | 39388(56.72) | 9709(35.36) | 2188(37.20) | 20036(51.66) | 15645(50.12) | 1779(45.62) | 88745(50.23) |
| Male | 20425(29.41) | 6686(24.35) | 973(16.54) | 13868(35.76) | 11187(35.84) | 1174(30.10) | 54313(30.74) |
| Missing | 9625(13.86) | 11066(40.30) | 2721(46.26) | 4879(12.58) | 4384(14.04) | 947(24.28) | 33622(19.03) |
| **Age(years)** | | | | | | | |
| < 18 | 687(0.99) | 518(1.89) | 13(0.22) | 1180(3.04) | 337(1.08) | 26(0.67) | 2761(1.56) |
| 18-44 | 5932(8.54) | 1382(5.03) | 237(4.03) | 4226(10.90) | 3135(10.04) | 433(11.10) | 15345(8.69) |
| 45–64 | 21527(31.00) | 3415(12.44) | 655(11.14) | 8936(23.04) | 7858(25.17) | 787(20.18) | 43178(24.44) |
| ≥65 | 16256(23.41) | 5486(19.98) | 636(10.81) | 12066(31.11) | 9177(29.40) | 1128(28.92) | 44749(25.33) |
| Not Specified | 25036(36.06) | 16660(60.67) | 4341(73.80) | 12375(31.91) | 10709(34.31) | 1526(39.13) | 70647(39.99) |
| **Weight(kg)** | | | | | | | |
| Mean(SD) | 81.75(25.27) | 73.54(25.83) | 76.45(20.12) | 76.61(27.77) | 75.19(23.42) | 71.20(21.76) | 78.28(25.74) |
| Median(Q1,Q3) | 79.40 (65.80,95.00) | 71.00 (58.96,86.17) | 73.92 (62.13,87.75) | 75.00 (62.80,90.00) | 73.00 (61.22,86.26) | 69.00 (57.15,82.55) | 76.25(63.40, 91.08) |
| N(Missing) | 22474(46964) | 4684(22777) | 1311(4571) | 12895(25888) | 7747(23469) | 1115(2785) | 50226(126454) |
| **Reporters** | | | | | | | |
| Consumer | 31069(44.74) | 11410(41.55) | 2386(40.56) | 14515(37.43) | 8307(26.61) | 667(17.10) | 68354(38.69) |
| Unspecified | 25061(36.09) | 1116(4.06) | 205(3.49) | 6644(17.13) | 765(2.45) | 180(4.62) | 33971(19.23) |
| Lawyer | 2387(3.44) | 5060(18.43) | 1747(29.70) | 784(2.02) | 6715(21.51) | 810(20.77) | 17503(9.91) |
| Physician | 6536(9.41) | 3310(12.05) | 665(11.31) | 6057(15.62) | 6099(19.54) | 991(25.41) | 23658(13.39) |
| Pharmacist | 2476(3.57) | 3216(11.71) | 332(5.64) | 5642(14.55) | 5089(16.30) | 787(20.18) | 17542(9.93) |
| Other health professional | 1909(2.75) | 3349(12.20) | 547(9.30) | 5141(13.26) | 4241(13.59) | 465(11.92) | 15652(8.86) |
| **Report countries** | | | | | | | |
| USA | 58669(84.49) | 18908(68.85) | 5434(92.38) | 19860(51.21) | 15967(51.15) | 1632(41.85) | 120470 (68.18) |
| France | 3983(5.74) | 2086(7.60) | 0(0.00) | 2152(5.55) | 4132(13.24) | 525(13.46) | 12878 (7.29) |
| Japan | 924(1.33) | 1342(4.89) | 0(0.00) | 457(1.18) | 2(0.01) | 440(11.28) | 3165 (1.79) |
| Canada | 757(1.09) | 227(0.83) | 109(1.85) | 550(1.42) | 2363(7.57) | 226(5.79) | 4232 (2.40) |
| United Kingdom | 701(1.01) | 2947(10.73) | 0(0.00) | 7731(19.93) | 444(1.42) | 62(1.59) | 11885 (6.73) |
| Other countries | 4404(6.34) | 1950(7.10) | 339(5.76) | 8033(20.71) | 8308(26.61) | 1015(26.03) | 24049 (13.61) |
| **Serious outcomes** | | | | | | | |
| Death | 3690(5.31) | 2468(8.99) | 499(8.48) | 1841(4.75) | 2137(6.85) | 370(9.49) | 11005(6.23) |
| Hospitalization-initial/ prolonged | 12376(17.82) | 5020(18.28) | 439(7.46) | 10396(26.81) | 8833(28.30) | 1135(29.10) | 38199(21.62) |
| Life-threatening | 1118(1.61) | 671(2.44) | 90(1.53) | 1936(4.99) | 1286(4.12) | 107(2.74) | 5208(2.95) |
| Disability | 1829(2.63) | 478(1.74) | 77(1.31) | 1573(4.06) | 538(1.72) | 89(2.28) | 4584(2.59) |
| Other serious outcomes | 33021(47.55) | 20574(74.92) | 3964(67.39) | 18955(48.87) | 21185(67.86) | 1717(44.02) | 99416(56.27) |

Esomeprazole demonstrates a distinct pattern compared to other PPIs. During short-term use (0–30 days), its adverse event reporting rate is lower than that of other PPIs, which may be attributed to its pharmacokinetic advantages as an isomer of Omeprazole. Studies indicate that Esomeprazole exhibits a higher area under the curve (AUC) with lower interindividual variability compared to Omeprazole, potentially contributing to a more stable

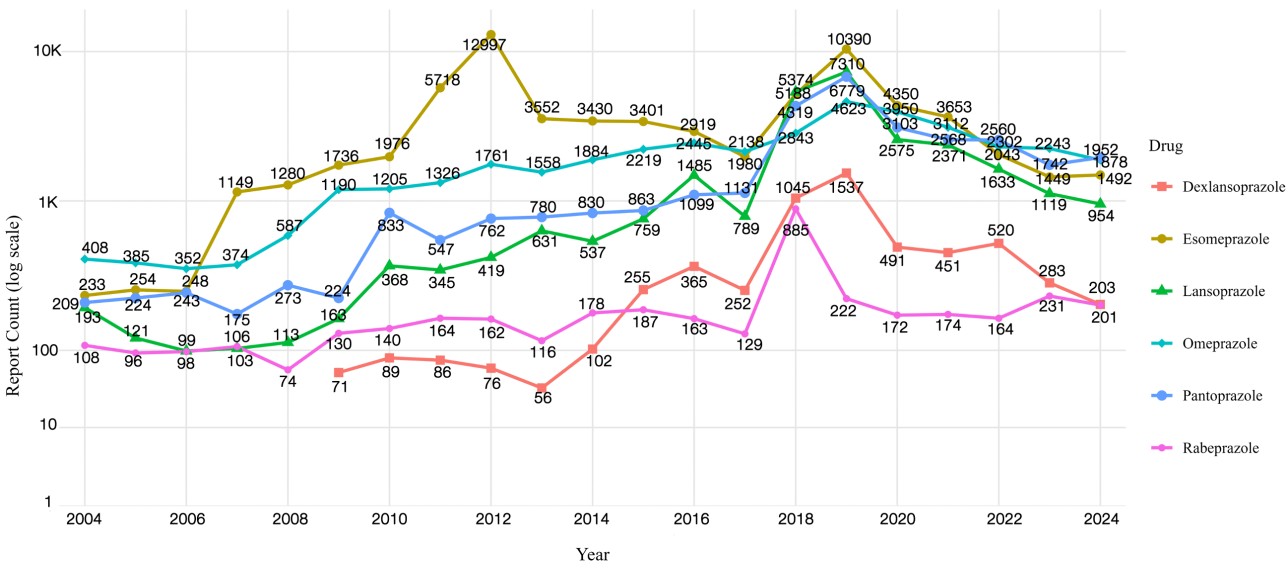

**Fig 1. PPI adverse reaction reports(2004-2024).** The x-axis shows the reporting year, and the y-axis displays the number of reports on a log10 scale (1, 10, 100, 1k, 10k), illustrating the yearly volume of adverse event submissions related to proton pump inhibitors.

acid-suppressive effect in the short term and consequently reducing the incidence of adverse reactions. Furthermore, Esomeprazole is primarily metabolized via CYP2C19 and CYP3A4, and its AUC increases significantly with prolonged administration (by 159% at a 40 mg dose), suggesting that extended high-level exposure may elevate the risk of drug-related adverse events [18]. Further research is required to elucidate the clinical implications of this phenomenon.

Table 2 presents the top 20 PTs of adverse events associated with six PPIs, along with the corresponding number of reported cases (N). To assess the association between PPIs and these adverse events, multiple signal detection methods were employed.The Reporting ROR was calculated based on disproportionality analysis, where a positive signal is identified if the lower bound of the 95% confidence interval (ROR025) exceeds 1. The PRR assesses the signal strength, considering an event a potential positive signal if PRR ≥ 2, χ² ≥ 4, and case count ≥ 3. The Information Component (IC), estimated using a Bayesian statistical approach, measures the deviation of observed event occurrence from expected values, and a positive signal is confirmed if the lower bound of the 95% confidence interval (IC025) exceeds 0. The EBGM is used to evaluate signal intensity, with a positive signal determined when EB05 (the lower 5% confidence bound of EBGM) exceeds 1.If a PT meets all four positive signal criteria simultaneously, the adverse event is classified as a PPI-related adverse event and is marked as "Y" in the "positive signal" column for identification.

According to Table 2, PPIs exhibit positive signals for multiple adverse events (AEs) in both the renal system and gastrointestinal system, with renal-related adverse events being particularly prominent. All six PPIs demonstrate statistically significant positive signals for CKD, AKI, end-stage renal disease (ESRD), and tubulointerstitial nephritis (TIN). In addition to reaffirming well-established adverse reactions such as acute interstitial nephritis, acute kidney injury, chronic kidney disease, and hypomagnesaemia, this study identified several emerging safety signals. Notably, significant signals were observed for hypertensive nephropathy and diabetic nephropathy, suggesting that PPI use may accelerate renal deterioration among patients with underlying cardiometabolic comorbidities. We also detected a clear signal for nephrogenic anaemia, an adverse reaction rarely reported in previous pharmacovigilance studies, warranting confirmation in prospective cohorts.

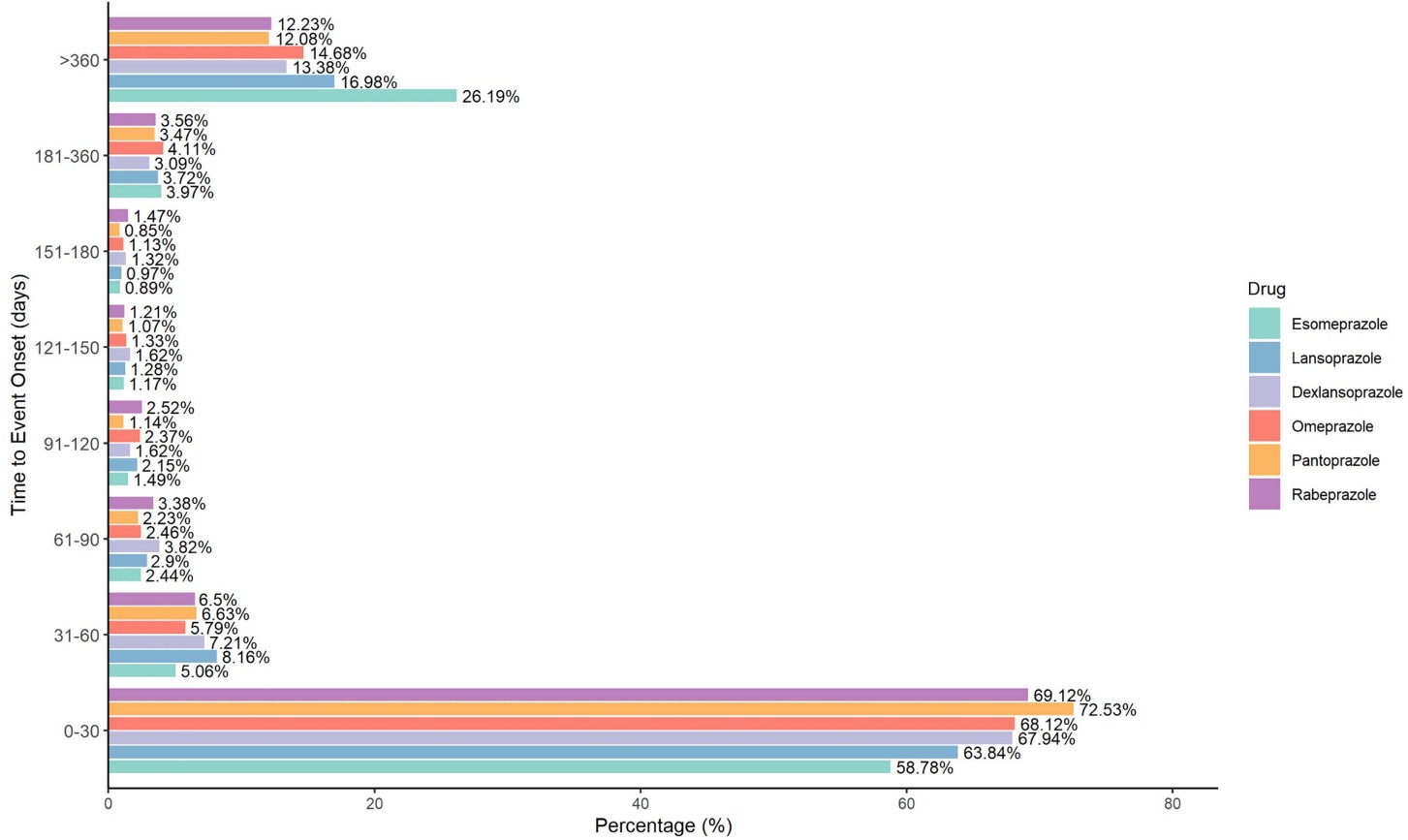

**Fig 2. Relationship between the number of adverse reaction reports related to PPIs and the duration of medication.** The x-axis lists individual PPIs, and the stacked bars represent the proportion of reports falling into different time-to-onset intervals (0–30 days, 31–90 days, 91–180 days, 181–360 days, > 360 days), allowing visual comparison of onset-time patterns across drugs.

Furthermore, prominent signals were identified for electrolyte disturbances, including hyponatraemia and hypocalcaemia, which, despite being supported mainly by isolated case reports, may carry important clinical implications. Importantly, our analysis quantified pharmacovigilance-level signals for rebound acid hypersecretion, rebound effect, and hyperchlorhydria. Although these phenomena have been clinically described, large-scale real-world evidence has been limited. Our findings suggest that acid overproduction following PPI discontinuation may be more common than previously recognized.

Among these, Lansoprazole exhibits the most pronounced positive signals for CKD (ROR = 45.80, IC025 = 5.03) and ESRD (ROR = 52.76, IC025 = 5.27), followed by Dexlansoprazole, which shows significant signals for CKD (ROR = 38.77, IC025 = 4.95) and ESRD (ROR = 38.84, IC025 = 4.97), indicating a potential risk of renal failure with long-term use. In addition, Lansoprazole also presents a notable positive signal for nephrogenic anaemia (ROR = 28.01, IC025 = 4.43). Notably, Lansoprazole demonstrates an exceptionally strong signal for hypertensive nephropathy (ROR = 240.21, IC025 = 6.36), which requires further clinical attention.

Meanwhile, Esomeprazole also exhibits an extremely strong positive signal for nephrogenic anaemia (ROR = 72.24, IC025 = 5.06), which is consistent with Wang's study results [19]. Furthermore, this drug shows the highest signal strength for rebound acid hypersecretion (ROR = 122.44, IC025 = 5.37), suggesting a significant risk of acid rebound upon discontinuation. However, this phenomenon has not been observed with Dexlansoprazole, Pantoprazole, or Rabeprazole.

                                    

**Table 2. The top 20 adverse reactions of six PPIs at the Preferred Term (PT) level.**

| | PT | N | ROR(95%CI) | PRR(χ2) | EBGM(95%CI) | IC025 | positive signal |
|---|---|---|---|---|---|---|---|
| Esomeprazole | | | | | | | |
| 1 | Chronic kidney disease | 16903 | 22.61(22.21-23.01) | 21.23(250392.43) | 16.48(16.20-16.78) | 4.02 | Y |
| 2 | Acute kidney injury | 9559 | 5.52(5.41-5.64) | 5.36(31683.98) | 5.04(4.94-5.15) | 2.30 | Y |
| 3 | Renal failure | 8896 | 5.45(5.33-5.57) | 5.30(29051.34) | 5.00(4.89-5.11) | 2.29 | Y |
| 4 | Gastrooesophageal reflux disease | 7594 | 8.57(8.36-8.78) | 8.35(44020.01) | 7.56(7.38-7.74) | 2.88 | Y |
| 5 | Renal injury | 5897 | 18.40(17.87-18.94) | 18.01(75316.02) | 14.50(14.09-14.93) | 3.81 | Y |
| 6 | End stage renal disease | 5113 | 29.80(28.83-30.80) | 29.25(98172.13) | 20.86(20.17-21.56) | 4.33 | Y |
| 7 | Dyspepsia | 4470 | 3.81(3.69-3.92) | 3.76(8626.37) | 3.62(3.51-3.73) | 1.81 | Y |
| 8 | Malaise | 4145 | 0.72(0.70-0.74) | 0.72(451.72) | 0.72(0.70-0.75) | −0.51 | |
| 9 | Pain | 2867 | 0.35(0.34-0.36) | 0.36(3414) | 0.36(0.35-0.37) | −1.53 | |
| 10 | Abdominal pain upper | 2575 | 0.99(0.95-1.03) | 0.99(0.17) | 0.99(0.95-1.03) | −0.06 | |
| 11 | Osteoporosis | 2469 | 4.97(4.77-5.17) | 4.93(7237.80) | 4.67(4.48-4.86) | 2.16 | Y |
| 12 | Vomiting | 2465 | 0.41(0.39-0.42) | 0.41(2104.04) | 0.41(0.40-0.43) | −1.33 | |
| 13 | Nephrogenic anaemia | 2285 | 72.24(68.13-76.60) | 71.63(78256.90) | 35.72(33.70-37.88) | 5.06 | Y |
| 14 | Nausea | 2132 | 0.20(0.19-0.21) | 0.21(6574.00) | 0.21(0.20-0.22) | −2.23 | |
| 15 | Abdominal discomfort | 2091 | 1.00(0.96-1.04) | 1.00(0.00) | 1.00(0.96-1.04) | −0.06 | |
| 16 | Tubulointerstitial nephritis | 2052 | 9.79(9.34-10.25) | 9.72(14086.89) | 8.65(8.25-9.05) | 3.04 | Y |
| 17 | Diarrhoea | 2014 | 0.24(0.23-0.25) | 0.25(4815.59) | 0.25(0.24-0.26) | −2.01 | |
| 18 | Rebound acid hypersecretion | 1918 | 122.44(113.67-131.90) | 121.57(83266.29) | 44.77(41.56-48.22) | 5.37 | Y |
| 19 | Dysphagia | 1762 | 1.48(1.41-1.55) | 1.47(266.00) | 1.47(1.40-1.54) | 0.48 | |
| 20 | Death | 1578 | 0.14(0.13-0.15) | 0.14(8423.85) | 0.14(0.14-0.15) | −2.86 | |
| Lansoprazole | | | | | | | |
| 1 | Chronic kidney disease | 12149 | 45.80(44.86-46.75) | 40.01(385667.51) | 33.43(32.75-34.13) | 5.03 | Y |
| 2 | Acute kidney injury | 5829 | 9.56(9.31-9.83) | 9.03(40106.82) | 8.68(8.45-8.92) | 3.08 | Y |
| 3 | Renal failure | 3649 | 6.13(5.92-6.33) | 5.93(14609.58) | 5.78(5.59-5.98) | 2.48 | Y |
| 4 | End stage renal disease | 3528 | 52.76(50.81-54.78) | 50.82(137141.37) | 40.61(39.11-42.17) | 5.27 | Y |
| 5 | Renal injury | 2954 | 23.45(22.56-24.37) | 22.74(55147.15) | 20.50(19.72-21.30) | 4.30 | Y |
| 6 | Tubulointerstitial nephritis | 1392 | 18.22(17.24-19.25) | 17.96(20456.25) | 16.55(15.66-17.49) | 3.95 | Y |
| 7 | Renal impairment | 1288 | 3.53(3.34-3.73) | 3.50(2266.11) | 3.45(3.27-3.65) | 1.70 | Y |
| 8 | Dialysis | 1054 | 17.51(16.43-18.65) | 17.32(14913.10) | 16.00(15.02-17.05) | 3.89 | Y |
| 9 | Gastrooesophageal reflux disease | 920 | 2.62(2.45-2.80) | 2.60(901.50) | 2.58(2.42-2.76) | 1.27 | Y |
| 10 | Death | 869 | 0.22(0.20-0.23) | 0.22(2428.82) | 0.23(0.21-0.24) | −2.25 | |
| 11 | Diarrhoea | 857 | 0.29(0.27-0.31) | 0.30(1474.91) | 0.30(0.28-0.32) | −1.85 | |
| 12 | Nephropathy | 851 | 21.48(20.00-23.06) | 21.30(14863.78) | 19.32(17.99-20.74) | 4.14 | Y |
| 13 | Hyperchlorhydria | 798 | 32.55(30.19-35.09) | 32.28(20794.36) | 27.88(25.86-30.06) | 4.64 | Y |
| 14 | Hyponatraemia | 682 | 2.69(2.49-2.90) | 2.68(708.58) | 2.65(2.46-2.86) | 1.29 | Y |
| 15 | Proteinuria | 619 | 7.77(7.17-8.42) | 7.73(3492.59) | 7.48(6.90-8.10) | 2.77 | Y |
| 16 | Hypertensive nephropathy | 570 | 240.21(212.57-271.44) | 238.76(61063.72) | 108.57(96.08-122.69) | 6.36 | Y |
| 17 | Nephrogenic anaemia | 556 | 28.01(25.62-30.61) | 27.85(12614.39) | 24.53(22.44-26.81) | 4.43 | Y |
| 18 | Diabetic nephropathy | 538 | 69.40(62.90-76.58) | 69.01(26718.57) | 51.39(46.57-56.70) | 5.41 | Y |
| 19 | Nephrolithiasis | 536 | 2.67(2.45-2.90) | 2.66(548.09) | 2.64(2.42-2.87) | 1.27 | Y |
| 20 | Rebound effect | 524 | 14.12(12.92-15.43) | 14.05(5930.89) | 13.18(12.06-14.40) | 3.56 | Y |
| Dexlansoprazole | | | | | | | |
| 1 | Chronic kidney disease | 2147 | 38.77(37.03-40.59) | 33.96(66889.72) | 32.98(31.50-34.52) | 4.95 | Y |
| 2 | Acute kidney injury | 886 | 7.76(7.25-8.31) | 7.41(4913.39) | 7.36(6.88-7.88) | 2.77 | Y |
| 3 | Renal failure | 748 | 6.91(6.42-7.44) | 6.65(3594.18) | 6.62(6.15-7.12) | 2.61 | Y |

*(Continued)*

**Table 2.** (Continued)

| | PT | N | ROR(95%CI) | PRR(χ2) | EBGM(95%CI) | IC025 | positive signal |
|---|---|---|---|---|---|---|---|
| 4 | End stage renal disease | 566 | 38.84(35.67-42.29) | 37.57(19503.55) | 36.37(33.40-39.60) | 4.97 | Y |
| 5 | Diarrhoea | 364 | 0.70(0.63-0.77) | 0.70(46.39) | 0.70(0.63-0.78) | −0.66 | |
| 6 | Renal injury | 319 | 12.70(11.36-14.20) | 12.48(3336.92) | 12.35(11.05-13.81) | 3.41 | Y |
| 7 | Death | 300 | 0.42(0.38-0.47) | 0.43(231.38) | 0.43(0.39-0.49) | −1.37 | |
| 8 | Gastrooesophageal reflux disease | 247 | 3.92(3.46-4.45) | 3.88(528.77) | 3.87(3.41-4.39) | 1.75 | Y |
| 9 | Nausea | 208 | 0.32(0.28-0.36) | 0.33(301.45) | 0.33(0.28-0.37) | −1.81 | |
| 10 | Tubulointerstitial nephritis | 172 | 11.63(10.00-13.52) | 11.52(1636.92) | 11.41(9.81-13.27) | 3.2 | Y |
| 11 | Abdominal pain upper | 164 | 0.99(0.85-1.16) | 0.99(0.00) | 0.99(0.85-1.16) | −0.23 | |
| 12 | Dizziness | 164 | 0.40(0.34-0.46) | 0.40(148.02) | 0.40(0.34-0.47) | −1.53 | |
| 13 | Headache | 145 | 0.28(0.24-0.33) | 0.28(270.56) | 0.28(0.24-0.33) | −2.05 | |
| 14 | Dyspnoea | 140 | 0.30(0.25-0.35) | 0.30(229.10) | 0.30(0.26-0.36) | −1.95 | |
| 15 | Vomiting | 138 | 0.36(0.30-0.42) | 0.37(154.93) | 0.37(0.31-0.43) | −1.69 | |
| 16 | Rebound acid hypersecretion | 129 | 49.87(41.77-59.52) | 49.49(5867.53) | 47.41(39.72-56.60) | 4.87 | Y |
| 17 | Abdominal pain | 119 | 0.63(0.53-0.75) | 0.63(25.67) | 0.63(0.53-0.76) | −0.92 | |
| 18 | Asthenia | 107 | 0.34(0.28-0.42) | 0.35(132.85) | 0.35(0.29-0.42) | −1.79 | |
| 19 | Pain | 104 | 0.20(0.17-0.24) | 0.21(329.46) | 0.20(0.17-0.25) | −2.55 | |
| 20 | Malaise | 96 | 0.26(0.21-0.31) | 0.26(201.37) | 0.26(0.22-0.32) | −2.20 | |
| Omeprazole | | | | | | | |
| 1 | Chronic kidney disease | 4096 | 8.25(7.99-8.52) | 8.04(23905.52) | 7.64(7.40-7.89) | 2.88 | Y |
| 2 | Acute kidney injury | 3204 | 3.34(3.23-3.46) | 3.29(5018.39) | 3.23(3.12-3.35) | 1.64 | Y |
| 3 | Renal failure | 2149 | 2.35(2.25-2.46) | 2.33(1616.98) | 2.31(2.21-2.41) | 1.14 | Y |
| 4 | Diarrhoea | 1972 | 0.45(0.43-0.47) | 0.46(1270.41) | 0.46(0.44-0.48) | −1.17 | |
| 5 | Nausea | 1782 | 0.33(0.31-0.34) | 0.34(2402.14) | 0.34(0.32-0.35) | −1.63 | |
| 6 | Pain | 1713 | 0.40(0.38-0.42) | 0.41(1500.28) | 0.41(0.39-0.43) | −1.35 | |
| 7 | Hypomagnesaemia | 1681 | 21.54(20.46-22.69) | 21.30(28058.74) | 18.50(17.57-19.49) | 4.12 | Y |
| 8 | Gastrooesophageal reflux disease | 1359 | 2.63(2.50-2.78) | 2.62(1338.46) | 2.59(2.45-2.73) | 1.29 | Y |
| 9 | Vomiting | 1321 | 0.42(0.40-0.44) | 0.42(1054.47) | 0.43(0.40-0.45) | −1.31 | |
| 10 | Dizziness | 1318 | 0.38(0.37-0.41) | 0.39(1267.29) | 0.39(0.37-0.42) | −1.42 | |
| 11 | Dyspepsia | 1290 | 2.02(1.91-2.14) | 2.01(649.16) | 2.00(1.89-2.11) | 0.91 | Y |
| 12 | Tubulointerstitial nephritis | 1262 | 11.01(10.39-11.66) | 10.92(10520.00) | 10.17(9.60-10.77) | 3.25 | Y |
| 13 | Abdominal pain upper | 1238 | 0.91(0.86-0.96) | 0.91(10.71) | 0.91(0.86-0.96) | −0.22 | |
| 14 | Headache | 1219 | 0.28(0.27-0.30) | 0.29(2205.77) | 0.29(0.27-0.31) | −1.87 | |
| 15 | Dyspnoea | 1208 | 0.31(0.29-0.33) | 0.32(1820.42) | 0.32(0.30-0.34) | −1.73 | |
| 16 | Hyperchlorhydria | 1104 | 32.47(30.40-34.68) | 32.21(26906.33) | 26.14(24.48-27.93) | 4.58 | Y |
| 17 | End stage renal disease | 1100 | 9.16(8.26-9.74) | 9.10(7431.57) | 8.58(8.07-9.13) | 3.00 | Y |
| 18 | Rebound effect | 1076 | 21.24(19.91-22.66) | 21.09(17784.93) | 18.34(17.20-19.57) | 4.08 | Y |
| 19 | Renal injury | 1055 | 5.14(4.84-5.47) | 5.11(3367.23) | 4.96(4.66-5.28) | 2.21 | Y |
| 20 | Hypocalcaemia | 1029 | 8.73(8.20-9.30) | 8.67(6562.89) | 8.20(7.70-8.74) | 2.93 | Y |
| Pantoprazole | | | | | | | |
| 1 | Chronic kidney disease | 7739 | 22.39(21.85-22.94) | 20.84(130990.03) | 18.71(18.26-19.17) | 4.19 | Y |
| 2 | Acute kidney injury | 4354 | 6.06(5.88-6.25) | 5.86(17081.60) | 5.70(5.52-5.88) | 2.46 | Y |
| 3 | Renal failure | 2348 | 3.37(3.23-3.51) | 3.31(3745.90) | 3.27(3.14-3.41) | 1.65 | Y |
| 4 | End stage renal disease | 2311 | 27.41(26.22-28.65) | 26.84(49825.88) | 23.37(22.36-24.43) | 4.47 | Y |
| 5 | Renal injury | 1226 | 7.84(7.40-8.30) | 7.76(6925.20) | 7.47(7.06-7.92) | 2.81 | Y |
| 6 | Nausea | 1186 | 0.28(0.27-0.30) | 0.29(2119.93) | 0.29(0.28-0.31) | −1.86 | |

*(Continued)*

 

**Table 2.** (Continued)

| | PT | N | ROR(95%CI) | PRR(χ2) | EBGM(95%CI) | IC025 | positive signal |
|---|---|---|---|---|---|---|---|
| 7 | Gastrooesophageal reflux disease | 1144 | 2.87(2.71-3.05) | 2.85(1359.93) | 2.82(2.66-2.99) | 1.41 | Y |
| 8 | Diarrhoea | 1127 | 0.34(0.32-0.36) | 0.34(1466.32) | 0.34(0.32-0.36) | −1.63 | |
| 9 | Tubulointerstitial nephritis | 1113 | 12.50(11.76-13.29) | 12.38(10879.69) | 11.62(10.94-12.36) | 3.44 | Y |
| 10 | Hypomagnesaemia | 917 | 14.17(13.25-15.17) | 14.06(10299.38) | 13.08(12.23-14.00) | 3.59 | Y |
| 11 | Dyspnoea | 916 | 0.31(0.29-0.33) | 0.31(1424.50) | 0.31(0.29-0.33) | −1.77 | |
| 12 | Death | 910 | 0.20(0.19-0.21) | 0.21(2896.65) | 0.21(0.19-0.22) | −2.37 | |
| 13 | Vomiting | 873 | 0.36(0.34-0.38) | 0.36(993.90) | 0.36(0.34-0.39) | −1.55 | |
| 14 | Fatigue | 786 | 0.19(0.18-0.20) | 0.20(2679.15) | 0.20(0.18-0.21) | −2.44 | |
| 15 | Dizziness | 784 | 0.30(0.28-0.32) | 0.30(1288.36) | 0.30(0.28-0.33) | −1.82 | |
| 16 | Headache | 714 | 0.21(0.20-0.23) | 0.22(2048.75) | 0.22(0.20-0.24) | −2.29 | |
| 17 | Malaise | 691 | 0.29(0.27-0.32) | 0.30(1165.67) | 0.30(0.28-0.32) | −1.85 | |
| 18 | Hypocalcaemia | 679 | 7.31(6.77-7.89) | 7.27(3526.21) | 7.01(6.50-7.58) | 2.68 | Y |
| 19 | Pruritus | 667 | 0.36(0.33-0.39) | 0.36(757.23) | 0.36(0.34-0.39) | −1.57 | |
| 20 | Abdominal pain | 664 | 0.55(0.51-0.59) | 0.55(239.78) | 0.55(0.51-0.60) | −0.96 | |
| Rabeprazole | | | | | | | |
| 1 | Chronic kidney disease | 487 | 12.54(11.44-13.74) | 12.00(4899.08) | 11.93(10.89-13.07) | 3.41 | Y |
| 2 | Renal failure | 475 | 6.99(6.38-7.67) | 6.72(2321.68) | 6.70(6.11-7.35) | 2.59 | Y |
| 3 | Acute kidney injury | 211 | 2.84(2.48-3.25) | 2.80(245.66) | 2.80(2.44-3.21) | 1.27 | Y |
| 4 | Pruritus | 138 | 0.76(0.64-0.90) | 0.76(10.27) | 0.76(0.65-0.90) | −0.63 | |
| 5 | Diarrhoea | 129 | 0.39(0.33-0.47) | 0.40(120.86) | 0.40(0.34-0.47) | −1.58 | |
| 6 | Rash | 116 | 0.53(0.45-0.64) | 0.34(46.51) | 0.54(0.45-0.65) | −1.15 | |
| 7 | Dizziness | 109 | 0.42(0.35-0.51) | 0.43(85.35) | 0.43(0.35-0.52) | −1.49 | |
| 8 | Nausea | 107 | 0.26(0.21-0.31) | 0.27(223.14) | 0.27(0.22-0.32) | −2.17 | |
| 9 | Headache | 102 | 0.31(0.26-0.38) | 0.32(153.56) | 0.32(0.26-0.39) | −1.93 | |
| 10 | Hypomagnesaemia | 91 | 13.35(10.86-16.43) | 13.2(1023.45) | 13.16(10.70-16.18) | 3.24 | Y |
| 11 | Hyponatraemia | 90 | 3.14(2.55-3.86) | 3.12(129.93) | 3.12(2.53-3.84) | 1.30 | Y |
| 12 | Tubulointerstitial nephritis | 88 | 9.43(7.64-11.64) | 9.36(654.00) | 9.31(7.55-11.49) | 2.78 | Y |
| 13 | Gastrooesophageal reflux disease | 86 | 2.16(1.75-2.67) | 2.15(53.18) | 2.15(1.74-2.66) | 0.77 | |
| 14 | Dyspnoea | 84 | 0.29(0.23-0.35) | 0.29(149.02) | 0.29(0.23-0.36) | −2.08 | |
| 15 | Abdominal pain | 80 | 0.68(0.54-0.84) | 0.68(12.33) | 0.68(0.54-0.85) | −0.88 | |
| 16 | Urticaria | 79 | 0.96(0.77-1.20) | 0.96(0.14) | 0.96(0.77-1.20) | −0.38 | |
| 17 | Vomiting | 79 | 0.33(0.26-0.41) | 0.33(107.09) | 0.33(0.27-0.42) | −1.89 | |
| 18 | Dyspepsia | 76 | 1.55(1.24-1.94) | 1.55(14.68) | 1.54(1.23-1.94) | 0.29 | |
| 19 | Malaise | 73 | 0.32(0.25-0.40) | 0.32(107.64) | 0.32(0.25-0.40) | −1.96 | |
| 20 | Arthralgia | 70 | 0.33(0.26-0.42) | 0.34(93.57) | 0.34(0.27-0.43) | −1.90 | |

In terms of electrolyte metabolism disorders, hypomagnesemia, hyponatremia, and hypocalcemia exhibit positive signals across different PPIs, indicating that long-term use may lead to electrolyte imbalances. Omeprazole (ROR = 21.54, IC025 = 4.12), Pantoprazole (ROR = 14.17, IC025 = 3.59), and Rabeprazole (ROR = 13.35, IC025 = 3.24) show strong positive signals for hypomagnesemia, which may be related to PPIs interfering with magnesium absorption and reabsorption in the small intestine and renal tubules. Hyponatremia is mainly observed with Lansoprazole (ROR = 2.69, IC025 = 1.29) and Rabeprazole (ROR = 3.14, IC025 = 1.30), suggesting that these drugs may affect renal water-electrolyte regulation, leading to decreased serum sodium levels. Hypocalcemia is primarily associated with Omeprazole (ROR = 8.73, IC025 = 2.93) and Pantoprazole (ROR = 7.31, IC025 = 2.68), indicating a potential reduction in calcium absorption due to

decreased gastric acid secretion, which may contribute to bone metabolism disorders, osteoporosis, and an increased risk of fractures. Therefore, for patients on long-term PPI therapy, particularly elderly individuals with impaired renal function, Regular monitoring of serum biochemical parameters and follow-up are recommended to minimize the potential risk of electrolyte disturbances.

All adverse events associated with PTs were categorized based on the SOC classification. After excluding the categories of Social Circumstances and Product Issues categories, the results were visualized using a heatmap. As illustrated in Figure (Fig 3), the distribution of PPI-related adverse events at the SOC level aligns with the findings at the PT level, predominantly affecting the Renal and Urinary Disorders and Gastrointestinal Disorders, followed by General Disorders and Administration Site Conditions and systemic metabolic abnormalities such as electrolyte Imbalance and systemic metabolic disorders. The use of PPIs in pregnant women appears relatively safe, aligning with the FDA labeling risk assessment and classification standards.

This study leveraged large-scale spontaneous reporting data from multiple countries and regions, providing strong external validity to the general population. The major safety signals detected—such as renal adverse events, electrolyte disturbances, and rebound acid hypersecretion—are supported by previous studies and well-established pharmacological

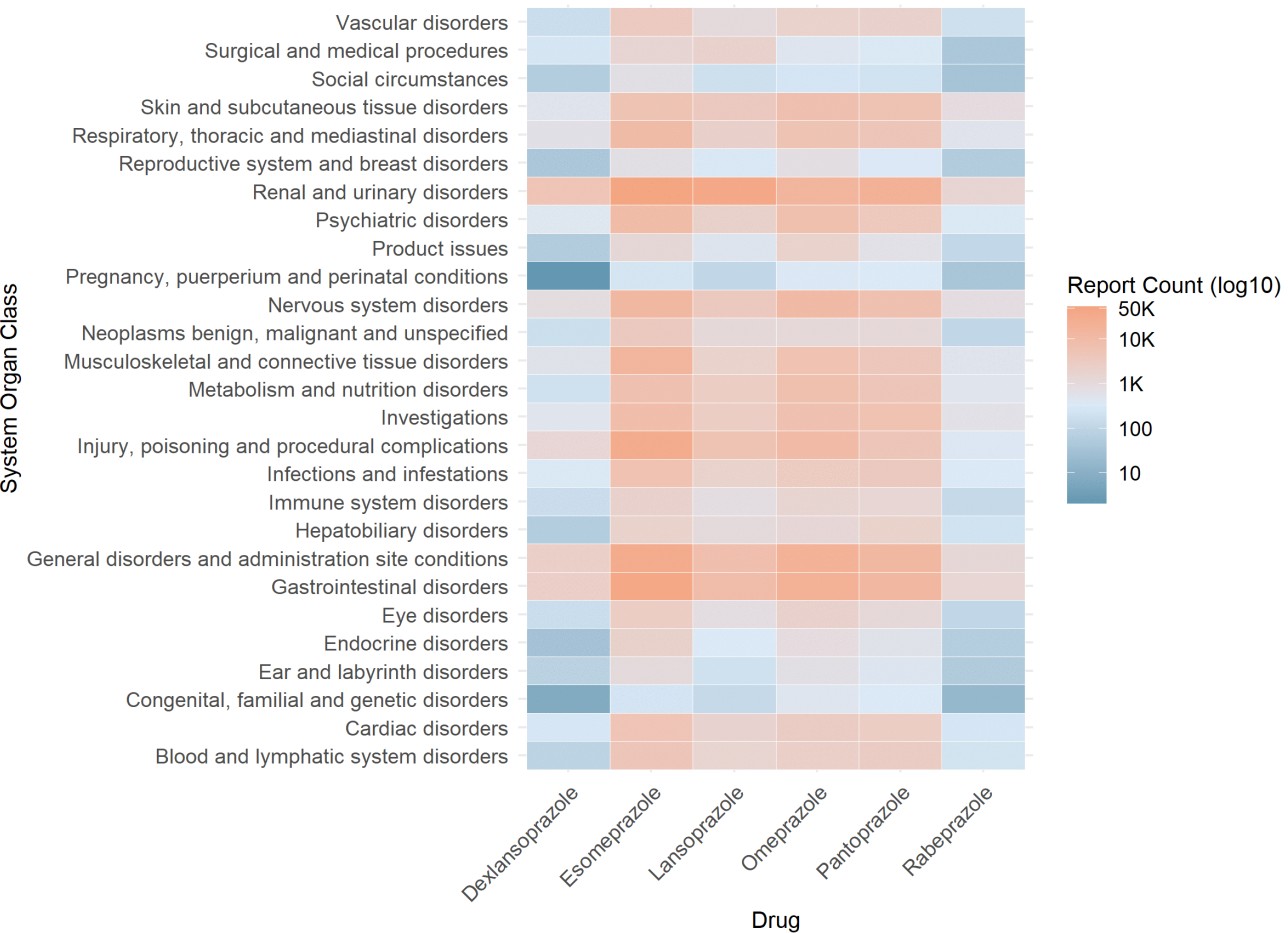

**Fig 3. Heatmap of the adverse reactions of six types of PPIs classified according to the System Organ Class (SOC).** The y-axis lists SOC categories, and the x-axis lists PPIs. Color intensity corresponds to the log10-transformed number of reports (per the legend scale), providing an overview of reporting distribution across organ systems.

mechanisms, indicating broad applicability. However, variations in prescribing preferences and drug-use patterns across regions limit the feasibility of detailed stratified analyses. Therefore, validation using domestic real-world databases, health insurance data, and prospective cohort studies remains essential to ensure the applicability of these findings to local clinical practice. Future high-quality studies are warranted to further strengthen the evidence base for these safety signals.

## 4. Discussion

PPIs have been associated with long-term off-label use, overuse, and prolonged administration in clinical practice [20], with their potential adverse effects drawing significant attention [9], particularly those affecting the renal/urinary and gastrointestinal systems. A prescription analysis study conducted across 45 hospitals in four Chinese cities revealed a substantial prevalence of inappropriate PPI prescriptions, especially among patients without clear indications for long-term excessive use [21]. An Italian study involving over 2000 patients from 27 nursing homes demonstrated that while 45.6% of patients were prescribed PPIs, only 50.7% of these prescriptions were based on evidence-based indications (such as peptic ulcers and Helicobacter pylori infection) [20]. Notably, the misuse of PPIs is particularly pronounced in the context of polypharmacy. These irrational prescribing patterns further exacerbate the risk of drug-related adverse events.

The findings of this study, in conjunction with existing evidence, necessitate heightened vigilance regarding the clinical use of PPIs, as long-term administration may significantly elevate the risk of adverse effects [22]. This concern is particularly evident in the renal and urinary systems, where significant associations between AKI or CKD events and PPIs have been reported since the early use of omeprazole, dating back to the 1990s [23,24]. Previous pharmacovigilance studies based on the FAERS database have shown significant signal associations between PPIs and various renal adverse outcomes. Jain et al. applied data-mining algorithms such as PRR and ROR, and identified marked disproportional reporting of acute kidney injury, chronic kidney disease, renal failure, and end-stage renal disease among PPI users, suggesting that the potential nephrotoxicity associated with PPI exposure warrants careful attention.An observational study shows a direct association between PPIs and CKD [25], which remains consistent even when baseline PPI users are compared directly with H2 receptor antagonist users [26]. Another study based on the FAERS database indicates that dexlansoprazole has the strongest association with both CKD and AKI among the six PPIs analyzed [27], and AKI results in more deaths, life-threatening events, hospitalizations, and disabilities compared to CKD. Our statistical analysis reveals strong associations for both lansoprazole and dexlansoprazole. Differences in data cleaning and filtering methods may account for variations, as some studies exclude non-professional reports but reach similar conclusions. A longitudinal follow-up study confirms a 50% increased risk of CKD development among PPI users versus non-users (HR = 1.50, 95% CI:1.14–1.96), with risk increasing proportionally to treatment duration [25]. Further research demonstrates that PPI users remain at elevated risk for CKD and end-stage renal disease even without experiencing AKI, suggesting that PPIs might exacerbate renal pathology through mechanisms independent of acute injury [22]. These findings argue against simplistic causal attributions and underscore the need for careful risk-benefit assessment in PPI therapy.

PPIs significantly interfere with the metabolism and absorption of ions such as calcium, magnesium, and iron by inhibiting gastric acid secretion and affecting renal excretion, thereby reducing their bioavailability. PPIs inhibit the $H^+/K^+$-ATPase in gastric parietal cells, markedly reducing gastric acid secretion and leading to decreased gastric acidity [28]. This low-acid environment diminishes the absorption of calcium, magnesium, and iron ions in the intestine. Additionally, PPIs can indirectly affect renal function by inhibiting gastrin secretion, which subsequently reduces the expression of parathyroid hormone-like hormone (PTHLH), further disrupting the metabolism of calcium, magnesium, and iron.

Gastric acid secretion facilitates calcium absorption. Prolonged use of PPIs can reduce the absorption of calcium ions from food, thereby decreasing bone mineral density and increasing the risk of osteoporosis and secondary fractures [28]. A large-scale study demonstrated that long-term PPI use [29], particularly with escalating doses or extended duration, is positively correlated with an increased risk of hip fractures, especially in men. Another study corroborated these concerns, prompting the FDA to include this warning in drug labels in 2010 [30].

Long-term PPI use has been causally linked to hypomagnesemia [31,32], with case reports and studies indicating that the specific mechanisms remain unclear. These mechanisms may involve increased cellular uptake of magnesium and reduced gastrointestinal absorption of magnesium, potentially leading to severe symptoms such as fatigue, muscle weakness, and arrhythmias [33]. Notably, the article further suggests that dietary interventions targeting the gut microbiota could serve as an effective treatment strategy [34]. Danziger et al [35]. demonstrated that the combined use of PPIs and diuretics increases the risk of hypomagnesemia in patients, necessitating long-term monitoring of serum magnesium levels in high-risk individuals and minimizing unnecessary polypharmacy.

Additionally, the lack of gastric acid can impair the dissolution and absorption of non-heme iron, and iron deficiency may further increase the risk of renal anemia [36]. Another study indicates that long-term PPI use, which suppresses gastric acid, is positively correlated with low vitamin B12 levels [37], thereby elevating the risk of related diseases. This study detected positive signals for renal anemia in both esomeprazole and lansoprazole, further emphasizing the importance of regular monitoring of relevant indicators during long-term PPI use to prevent associated complications.

During the treatment of gastric acid suppression with PPIs, instances of acid rebound and unresolved symptoms of GERD may occur. Although the use of PPIs in GERD treatment is widely recognized, their efficacy in refractory GERD is limited. Studies suggest that this refractoriness may be attributed to multiple factors [38], including rapid PPI metabolism, poor medication adherence, nocturnal acid breakthrough, non-acidic reflux, or functional heartburn. It is essential to first evaluate the appropriateness of PPI use, further clarify the underlying etiology, and adjust the treatment plan accordingly. Additionally, gradual dose reduction during discontinuation is crucial to avoid acid rebound caused by feedback stimulation of gastric parietal cells, which may lead to symptoms such as heartburn, acid regurgitation, and dyspepsia [39].

In summary, PPIs should be prescribed or sold only for clear indications, at low doses, and for short durations. PPI treatment should be discontinued in users without indications or with unclear indications, and even those with indications should avoid long-term use unless necessary. Patients with primary kidney disease should exercise particular caution when using PPIs, and long-term users should undergo regular renal function monitoring [40]. Long-term PPI use without indications should be avoided, and dose reduction should be considered when appropriate to mitigate potential mortality risks [41].

In our database research, rabeprazole exhibited the lowest number of adverse reaction reports but the highest mortality rate, whereas esomeprazole had the highest number of cases but a lower mortality rate. This phenomenon suggests the potential existence of drug selection bias: high-risk patients may be more inclined to use specific PPIs, such as rabeprazole for end-stage patients. A study indicated that rabeprazole does not require dose adjustment for patients with stable, end-stage renal failure; however, due to the small sample size, further clinical trials are still needed [42].

Bias exists in the information collection of databases, where reports on specific drugs or biological products cannot confirm whether adverse reactions are caused by the drug in question; they merely represent the reporter's PS [43]. For any given adverse drug event (ADE) report, the ADE signal calculated based on disproportionality analysis indicates a statistical correlation between the target drug and the target ADE, but not a biological one. This does not prove a definitive causal relationship, necessitating further in-depth investigation through clinical trials and prospective real-world studies. Additionally, the regional distribution of FAERS is predominantly centered in the United States, with limited data from Asia and Africa, resulting in insufficient representativeness. Given the variation in prescribing habits and drug-use preferences across different countries and regions, fine-grained stratified analyses based solely on spontaneous reports are difficult to achieve. The reporting of case data is primarily consumer-driven, with voluntary and self-reported submissions lacking professionalism and review, inevitably leading to issues such as underreporting, overreporting, inconsistent or missing information [44]. The high proportion of consumer-submitted reports may further introduce bias, as consumers tend to report more subjective or symptom-driven reactions, while clinically silent or laboratory-based events are less likely to be captured. In addition, consumers may be more influenced by media attention or personal perception of risk, which can

selectively amplify certain adverse events. The limitations of database analysis underscore the need for more clinical research to validate the robustness and clinical significance of the observed signals.

## 5. Conclusions

This study systematically analyzed PPI-associated adverse events using data from the FAERS database, aiming to assess potential risks and associations, thereby providing scientific evidence and theoretical support for clinical practice.

## Supporting information

**S1 Table. SOC-level adverse reactions of six PPIs.**
(XLSX)

**S2 Table. PPI adverse reaction reports(2004–2024).**
(XLSX)

**S3 Table. Adverse reaction dataset stratified by time.**
(XLSX)

## Acknowledgments

We would like to express our gratitude to the Department of Pulmonary and Critical Care Medi-cine at the Regional Medical Center for National Institute of Respiratory Disease, Sir Run Run Shaw Hospital, for their support of this manuscript.

## Author contributions

**Conceptualization:** Zhenyu Wang, Jianan Jin, Guimei Wang, Hanliang Jiang.

**Data curation:** Zhenyu Wang, Jianan Jin.

**Formal analysis:** Zhenyu Wang, Jianan Jin, Guimei Wang.

**Funding acquisition:** Guoqi Zhou, Hanliang Jiang.

**Investigation:** Guoqi Zhou, Hanliang Jiang.

**Project administration:** Zhenyu Wang.

**Resources:** Zhenyu Wang, Jianan Jin.

**Software:** Zhenyu Wang, Jianan Jin.

**Supervision:** Guoqi Zhou, Hanliang Jiang.

**Visualization:** Zhenyu Wang, Jianan Jin, Guimei Wang.

**Writing – original draft:** Zhenyu Wang, Jianan Jin, Guimei Wang.

**Writing – review & editing:** Guoqi Zhou, Hanliang Jiang.

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
