## [Decision Letter · Decision Letter 0]

26 Nov 2025

Dear Dr. Jiang,

Thank you for submitting your manuscript to PLOS ONE. After careful consideration, we feel that it has merit but does not fully meet PLOS ONE’s publication criteria as it currently stands. Therefore, we invite you to submit a revised version of the manuscript that addresses the points raised during the review process.

We look forward to receiving your revised manuscript.

Kind regards,

Sara Mucherino

Academic Editor

PLOS ONE

Journal Requirements:

3. We note that your Data Availability Statement is currently as follows:

“All relevant data are within the manuscript and its Supporting Information files.”

5. Please upload a copy of Supporting information figures (S2, S3, S5) and tables (S1, S4) which you refer to in your text on pages 17.

**Additional Editor Comments:**

Dear Dr Hanliang Jiang, please, where possible, address the comments of reviewers below when reviewing your manuscript.

Thanks

Reviewers' comments:

Reviewer's Responses to Questions

**Comments to the Author**

1. Is the manuscript technically sound, and do the data support the conclusions?

Reviewer #1: No

Reviewer #2: Yes

2. Has the statistical analysis been performed appropriately and rigorously?

Reviewer #1: Yes

Reviewer #2: Yes

3. Have the authors made all data underlying the findings in their manuscript fully available?

Reviewer #1: Yes

Reviewer #2: Yes

4. Is the manuscript presented in an intelligible fashion and written in standard English?

Reviewer #1: No

Reviewer #2: Yes

Reviewer #1: 1. Provide an outline of the entire study design, including primary and sensitivity analyses performed, and other designs such as case-by-case analysis or literature review.

2. Explain more in detail: Data description, access, and pre-processing. Variables definition. Statistical methods.

3. Case-by-case analysis: Presenting the case-by-case analysis of key variables. Presenting the causality assessment, if applicable

4. Clearly discriminate between expected reactions and emerging safety signals.

Discuss the external validity of the results to the general population or in their country

5. Limitations including notification by consumer, which represents a high percentage and can have a great association bias

6. Author has missed out important studies like Adverse effects of proton pump inhibitors (PPIs) on Renal system using Data Mining Algorithms (DMAs). Expert Opinion on Drug Safety

7. How data was cleaned?

Reviewer #2: I was pleased to review this important research. At a time when the overuse of proton pump inhibitors is on the rise, this study sends alarming signals that healthcare practitioners should take seriously. I have a few suggestions for improving the article, which are as follows:

1. There is repeating of references e.g., the reference (20) is the same reference (22); also, the reference (26) is the same reference (28).

2. There is a discrepancy regarding financing in the "Financial Disclosure" section and under the "Funding" section.

3. In Table 1, I suggest adding a column on the right showing the "Total Number and Percentage" for each row. Also in Table 1, it would be preferable for the row indicating "missing" values of weight, to follow the mean and median in order.

4. The figures are not entirely clear, especially figure number (1).

5. In the discussion section it was stated that "A study indicated that rabeprazole does not require dose adjustment for patients with stable, end-stage renal failure..." Please cite the reference.

6. In conclusion, it should be noted that the study demonstrated a strong correlation between the use of proton pump inhibitors and the occurrence of serious and multiple side effects.

**Do you want your identity to be public for this peer review?** For information about this choice, including consent withdrawal, please see our Privacy Policy

Reviewer #1: **Yes:** anoop kumar

Reviewer #2: **Yes:** ABDULRAZZAQ YAHYA AHMED AL-KHAZZAN

---

## [Author Response · Author response to Decision Letter 1]

2 Dec 2025

Dear Editor,

Thank you very much for your time and for coordinating the review process. We sincerely appreciate the valuable and constructive comments provided by both the editor and reviewers, which have greatly improved the quality of our manuscript.

In accordance with the editorial requirements, we have carefully revised the manuscript and uploaded all requested files. Please find the updated materials below:

1. Financial Disclosure Statement

We have updated the financial disclosure to:

“This work was supported by the National Natural Science Foundation of China (Grant No. 62176230) and the Zhejiang Medical and Health Science and Technology Project (Grant No. 2022KY827). The funders had no role in study design, data collection and analysis, decision to publish, or preparation of the manuscript.”

The same update has been made in the submission system, and we kindly ask the editor to review and confirm the change.

2. Data File for Figures

This study is based entirely on publicly accessible data from the FDA FAERS database. All raw data were obtained from this public source, and the data-cleaning procedures have been fully described in the Methods section. After processing the FAERS data using R, the final analytical dataset used for statistical calculations and figure generation was exported to Excel. We have therefore uploaded this final dataset as Supporting Information, as it contains all values necessary to reproduce the results presented in the manuscript.

3. ORCID iD Update

The ORCID iD of the corresponding author has been successfully linked as required.

4. Supporting Information Files

Thank you for pointing out the issue regarding the Supporting Information files (S1–S5). Upon re-evaluation, we realized that our team had previously misunderstood the requirements for Supporting Information. All key figures and data have now been fully integrated into the main manuscript, and no citations to Supporting Information remain in the text; therefore, the original S1–S5 files have been removed in the revised submission. In accordance with the Data Availability Statement requirements, we have uploaded the final analytical dataset (Excel file) used for statistical calculations and figure generation as Supporting Information. If the editor considers any additional supplementary materials necessary, we would be happy to provide them. Thank you again for your guidance and support.

Sincerely,

Hanliang Jiang

Reviewer #1

Comment 1

Provide an outline of the entire study design, including primary and sensitivity analyses performed, and other designs such as case-by-case analysis or literature review.

Response:

Thank you very much for this insightful comment. We have revised and expanded the Methods section (Section 2.1: Data Source) to more clearly outline the overall study design, including the data source, study scope, and key analytic components. We also describe how potential confounding factors—such as combination therapies and clearly non–drug-related adverse events—were excluded to enhance signal accuracy, and we interpret the main findings in the Discussion with reference to previous literature on PPI-related renal risk. We fully acknowledge the importance of sensitivity analyses; however, due to recent structural updates of the FAERS database and the substantial workload required for multiple subgroup analyses, these were not feasible in the current study. This has been explicitly noted as a limitation and identified as a priority for future work.

Comment 2

Explain more in detail: Data description, access, and pre-processing. Variables definition. Statistical methods.

Response:

Thank you for raising this important point. We have revised the Materials and Methods section to provide more detailed information on data description, access, preprocessing and cleaning procedures, variable definitions, and the statistical methods used. Specifically, we now report the time frame of data extraction, deduplication based on FDA guidance, exclusion and filtering criteria, and independent verification of data processing. For disproportionality analysis (e.g., ROR and PRR), we refer to these as well-established methods in pharmacovigilance and therefore did not reproduce the standard formulas in the main text, but we would be pleased to add them if the reviewer or editor considers it necessary.

Comment 3

Case-by-case analysis: Presenting the case-by-case analysis of key variables. Presenting the causality assessment, if applicable.

Response:

Thank you for this suggestion. We considered conducting case-by-case analyses; however, due to the spontaneous nature of the FAERS database and the frequent lack of detailed clinical, temporal, and dosing information, a robust causality assessment at the individual case level was not feasible. For this reason, we did not perform case-level analysis in the present study, and we have clarified this point in the revised manuscript.

Comment 4

Clearly discriminate between expected reactions and emerging safety signals. Discuss the external validity of the results to the general population or in their country.

Response:

Thank you for the valuable suggestion. We have revised the Results section to clearly distinguish between adverse reactions that are already expected or labelled and those that may represent emerging safety signals based on disproportionality patterns. In addition, we have expanded the Discussion to address the external validity of our findings, both in relation to the general population and to their applicability in specific national or regional contexts. We also further discuss limitations related to data representativeness and spontaneous reporting bias.

Comment 5

Limitations including notification by consumer, which represents a high percentage and can have a great association bias.

Response:

Thank you for this helpful comment. We have added a specific paragraph to the Discussion addressing the potential reporting bias introduced by the high proportion of consumer-submitted reports in FAERS, including how this may influence the strength and direction of the observed associations.

Comment 6

Author has missed out important studies like “Adverse effects of proton pump inhibitors (PPIs) on Renal system using Data Mining Algorithms (DMAs)” in Expert Opinion on Drug Safety.

Response:

Thank you for pointing this out. We agree that this is an important related study. We have now cited the article “Adverse effects of proton pump inhibitors on renal system using data mining algorithms” (Expert Opinion on Drug Safety) in the revised Discussion and briefly compared our findings with theirs to better contextualize our results.

Comment 7

How data was cleaned?

Response:

Thank you for the suggestion. We have added more detail on data cleaning to the Methods section (Section 2.1: Data Source), including the deduplication procedure following FDA recommendations, the exclusion of duplicate or conflicting records, and the removal of reports with missing or implausible values in key variables such as age, sex, and weight. These steps are now explicitly described in the revised manuscript.

Reviewer #2

Comment 1

There is repeating of references e.g., the reference (20) is the same reference (22); also, the reference (26) is the same reference (28).

Response:

Thank you for pointing this out. The duplicate references have been carefully reviewed and corrected in the revised manuscript to ensure accuracy and consistency.

Comment 2

There is a discrepancy regarding financing in the "Financial Disclosure" section and under the "Funding" section.

Response:

Thank you for your careful observation. The funding information has been reviewed, synchronized, and corrected in both sections to maintain consistency.

Comment 3

In Table 1, I suggest adding a column on the right showing the "Total Number and Percentage" for each row. Also in Table 1, it would be preferable for the row indicating "missing" values of weight, to follow the mean and median in order.

Response:

Thank you for the helpful suggestion. We have revised Table 1 by adding a “Total Number and Percentage” column, and repositioned the row for “missing” weight values to follow the mean and median rows for improved readability.

Comment 4

The figures are not entirely clear, especially figure number (1).

Response:

Thank you for this important comment. We have updated Figure 1 with a higher-resolution version, optimized its layout, and enhanced labeling. We also added clearer annotations and expanded the figure legend to improve interpretability for readers.

Comment 5

In the discussion section it was stated that "A study indicated that rabeprazole does not require dose adjustment for patients with stable, end-stage renal failure..." Please cite the reference.

Response:

Thank you for pointing this out. We have added the appropriate reference to support the statement regarding rabeprazole dosing in patients with stable end-stage renal failure.

Comment 6

In conclusion, it should be noted that the study demonstrated a strong correlation between the use of proton pump inhibitors and the occurrence of serious and multiple side effects.

Response:

Thank you for the suggestion. We have revised the Conclusion section to emphasize the observed strong correlation between proton pump inhibitor use and the occurrence of serious and multiple adverse effects, as recommended.

---

## [Decision Letter · Decision Letter 1]

26 Dec 2025

Systematic Analysis of Proton Pump Inhibitors-Related Adverse Reactions Using the FDA Adverse Event Reporting System Database

PONE-D-25-21825R1

Dear Dr. Jiang,

We’re pleased to inform you that your manuscript has been judged scientifically suitable for publication and will be formally accepted for publication once it meets all outstanding technical requirements.

Kind regards,

Sara Mucherino

Academic Editor

PLOS One

Reviewers' comments:

Reviewer's Responses to Questions

**Comments to the Author**

Reviewer #1: All comments have been addressed

Reviewer #2: All comments have been addressed

2. Is the manuscript technically sound, and do the data support the conclusions?

Reviewer #1: Partly

Reviewer #2: Yes

3. Has the statistical analysis been performed appropriately and rigorously?

Reviewer #1: Yes

Reviewer #2: Yes

4. Have the authors made all data underlying the findings in their manuscript fully available?

Reviewer #1: No

Reviewer #2: Yes

5. Is the manuscript presented in an intelligible fashion and written in standard English?

Reviewer #1: Yes

Reviewer #2: Yes

Reviewer #1: Author has addressed most of my comments in the revised manuscript. Therefore, no further comments.

Reviewer #2: (No Response)

**Do you want your identity to be public for this peer review?** For information about this choice, including consent withdrawal, please see our Privacy Policy

Reviewer #1: **Yes:** anoop kumar

Reviewer #2: **Yes:** ABDULRAZZAQ YAHYA AHMED AL-KHAZZAN

---

## [Editor Report · Acceptance letter]

PONE-D-25-21825R1

PLOS One

Dear Dr. Jiang,

I'm pleased to inform you that your manuscript has been deemed suitable for publication in PLOS One. Congratulations! Your manuscript is now being handed over to our production team.

Kind regards,

on behalf of

Dr. Sara Mucherino

Academic Editor

PLOS One